# International testing and refinement of AI algorithms predicting acute leukemia subtypes from routine laboratory data

Amin T. Turki [1,2,3] ✉, Yi Fan[4], Alberto Hernández-Sánchez [5], Wellington Silva[6], Shaun Fleming[7], Koray Yalcin[8], Catharina H.M.J. Van Elssen[9], Yazan Madanat [10], Magdalena Karasek [11,12], Mahmoud Aljurf[13], Matteo G. Della Porta [14], Alexandra Martinez-Roca [15], Luca Guarnera[16], Katarina Steffen[17], Evangelia Antoniou[18], Maria M. Rivas [19], Deepak K. Mishra[20], Ansgar T. Blum[2], Stephania Niry Manantsoa[21], Adeniyi Adiat[22], Amir Enshaei [23], Felicitas Thol [24], Maria Teresa Voso [16], Jia Chen[4], Tusneem Ahmed Elhassan[13,25], Anthony V. Moorman [23], María Belén Vidriales[5], Nina R. Neuendorff[2], Ahmet Koc[26], Pratyush Mishra[20], Dirk Strumberg[2], Roma S. Fourmanov[9], Lukas Heine[3], Jens Kleesiek [3], Daniel Munárriz[15], Gianluca Asti[14], Mridula Mokoonlall [7], Marisa Kometas[10], Eduardo Rego[6], Rabea Mecklenbrauck [24], Marta Sobas[11,12], Depei Wu [4], Felix Nensa [3] & Merlin Engelke [3]

Despite advances for patients with acute leukemia health disparities limit access to diagnosis and treatment. Artificial Intelligence (AI) approaches may address some disparities. We retrospectively assemble a diverse, international cohort of 6206 leukemia patients from 20 centers to test an AI tool designed to support leukemia diagnosis using standard laboratory results. Executing the pretrained algorithm results in varying accuracy metrics. With confidence cutoff predictions, 2000-fold bootstrapped area under the curve (AUROC) metrics are 0.94 for acute myeloid leukemia (AML), 0.98 for the promyelocytic subtype and 0.84 for acute lymphoblastic leukemia. However, this cutoff excludes 70.8–92.5% of patients from predictions. We improve accuracy and robustness, while maintaining generalizability via an ensemble of Isolation Forest and Local Outlier Factor increasing AUROC for AML from 0.72 to 0.84 (hold-out test set, patients below confidence threshold), while excluding only 12.1% of patients. Furthermore, we retrain the algorithm for pediatric patients.

While the care for patients with acute myeloid leukemia (AML) and acute lymphoblastic leukemia (ALL) has been broadly improved over the past decades e.g., via novel and targeted treatment options[1–3], refined genomic risk stratification[4] and both reduced early mortality[5–7], and reduced nonrelapse mortality after HCT[8,9], barriers to equitable diagnosis and treatment prevail and limit the access to these benefits[10]. In particular, patients in low- and middle-income countries face

significant barriers to accessing specialized diagnosis and care[11,12], but systemic health disparities are also observed within high-income countries[13]. Several constraints, including access to flow cytometry and molecular genetics testing[14], may challenge a timely diagnosis of acute leukemias and delay referral or specialized care. For patients with acute promyelocytic leukemia (APL), this diagnostic gap significantly increases early mortality[15,16]. APL has favorable long-term outcomes,

with > 90% long-term remission in patients who survive the critical first 4–6 weeks until hematological remission. Early diagnosis and treatment, which combines ATRA, cytoreduction, and efficient coagulation management, aims to reduce morbidity and mortality due to thromboembolic and bleeding events and differentiation syndrome. Early death rates still range from 5–10% in clinical trials and up to 30% in real-world data from middle-income countries[15] and tragically often occur before complete APL diagnosis. In addition, for other subtypes of acute leukemia, facilitated diagnosis is warranted to allow early referral to specialized centers or initiation of targeted therapy under restricted resources as early as possible and manage acute leukemia patients appropriately to avoid early death.

The potential of Artificial Intelligence (AI) with deep learning approaches using marrow morphology images to support the diagnosis of acute leukemias has been previously demonstrated with high levels of accuracy[17,18]; however, at present, such approaches require complex infrastructures and substantial resources that limit access. Recently, routine laboratory features have been leveraged to develop and test machine learning (ML) classification algorithms for predicting three types of leukemia (AML, ALL, and APL) in multicenter French cohorts[19], an approach that could make AI diagnostic support broadly available at low costs. Recognizing the importance of independent international validation and real-world evaluation of potentially relevant models, we here assemble a large, global cohort to systematically test this AI classifier's accuracy, generalizability, speed, and robustness and to refine this model for practical clinical application.

## Results

### Testing on a diverse cohort with highly accurate predictions for AML, yet not applicable to all patients

This global cohort ($n = 6206$) was assembled from 20 contributing centers in 16 countries across 5 continents (Fig. 1a). Its characteristics are detailed in Supplementary Table 3. The cohort was diverse (geographically, socially, for age (range 0.08–97 years)), including both sexes (male 50%), adults (≥ 18 years, $n = 4460$), and pediatric patients ($n = 1746$), which were analyzed separately. The ethnicity was not recorded, but partly associates with the site geography. According to the UN classifications, 43% (7/16) of the participating countries were of low- and middle-income (Fig. 1b and Supplementary Table 2). AML was the most common diagnosis ($n = 3510$, Fig. 1c). In adults, the pretrained model's 2000-fold bootstrap AUROC was 0.82 (95% CI, 0.82–0.82) for the prediction of AML, 0.92 (95% CI, 0.92–0.92) for APL and 0.79 (95% CI, 0.79–0.79) for ALL (Fig. 1d). Given that XGBoost models reflect the confidence of its predictions, we tested the previously published confidence cutoff[19]. These predictions reached in specific sites very high AUROCs of up to 1.0 for AML, 1.0 for APL, and 0.85 for ALL (Supplementary Data 1 and Fig. 1e). For clinical use, however, this cutoff was overly restrictive, excluding 70.8–92.5% of adult patients from predictions, which was why we focused on the performance of the general model. Here, high-accuracy metrics were obtained across geographic regions (Fig. 2 and Supplementary Fig. 3a), yet with substantial variance between centers and classes. Bootstrapped AUROC for APL ranged from 0.98 for Salamanca, Spain, 0.93 for Suzhou, China, and 0.63 for Antananarivo, Madagascar. AUROC for adult AML reached 0.93 (95% CI, 0.93–0.94) in Melbourne, Australia, and Maastricht, the Netherlands (Fig. 2). The highest F1 scores (range 0.56–0.95) were obtained with AML, whereas ALL (0.15–0.70) and APL F1 scores were lower due to a precision-recall tradeoff (Supplementary Fig. 3a). Interestingly, the probability distribution of true positives per country per class of the adult AML model was relatively high (i.e., approximately 0.9) across countries (Fig. 3a). Its variance was comparatively small, with some spread in the variance of smaller-sized cohorts. For adult ALL and APL, the variation between centers was substantial, yet the number of cases was also much smaller. The probability distribution of the false negative values was lowest in APL, supporting the

favorable results regarding the AUROC (Supplementary Fig. 3b). Given the observed variations in performance across cohorts and classes, we also compared our cohorts' feature distributions to those French cohorts, in which the pretrained model was developed and validated (Supplementary Data 2). In APL, there were differences in fibrinogen levels (mean 1.84 g/l vs. 2.51 g/l in French cohorts) and in ALL, monocyte and lymphocyte counts were substantially higher (mean 3.90/nl vs. 0.49/nl and 18.62/nl vs. 3.72/nl).

### Differences in feature distributions between acute leukemia subtypes. Coagulation parameters and MCV are most relevant for the model

Explainable predictions may support the clinical implementation of AI models in addition to their accuracy, robustness, and generalizability. We comparatively analyzed the range of laboratory results of correctly classified and misclassified adult patients within the leukemia subtype algorithm. Among correctly classified patients with APL, those with MCHC values above 340 g/l, lymphocyte counts below 1 g/l, prothrombin times below 75%, and fibrinogen levels below 2 g/l were enriched (Fig. 3b). For patients with AML, an MCV above 90 fl and higher monocyte counts support this prediction. The majority of correctly predicted ALL patients had a PT above 85% and an MCV below 90 fl. However, the algorithm was susceptible to high monocyte counts, which are regularly classified as AML, when they occur in other classes (Supplementary Fig. 4). In adults, feature importance analysis revealed that fibrinogen (270.80), age (260.52), and MCV (170.04) were the most discriminative variables. For AML, SHAP analysis revealed that monocytes (0.284 for %; 0.165 for counts; Fig. 3c) were the strongest positive predictor, followed by MCV (0.133). In ALL patients, the monocyte percentage (− 0.229) and PT percentage (− 0.158) were the strongest negative predictors, whereas the MCHC (0.036) showed modest positive feature importance. Low fibrinogen levels (− 0.361) and low MCHCs (− 0.141) favor the prediction of APL. According to the subgroup analyses, the algorithm generalized well between men (AUROC AML 0.82, APL 0.92 and ALL 0.8) and women (AUROC AML 0.8, APL 0.93 and ALL 0.78). Patients with leukocytosis and peripheral blasts (WBC > 20,000/μl) had predictions comparable to those of the entire population (AUROC AML 0.8, APL 0.93 and ALL 0.76). When we analyzed AML patients with documented ELN risk status from high-income countries, we observed comparable performance between genetic risk profiles (AUROC 0.91 for ELN favorable risk AML, 0.9 for intermediate-risk AML, and 0.92 for adverse risk AML).

### Refining the algorithm predictions for adults to achieve a good generalizability

Given the observed shortcomings, we applied several methods to refine the algorithm's predictions for greater generalizability and clinical applicability. All adult data points are plotted in Supplementary Fig. 5. First, we applied dimensionality reduction with PCA to the features and successfully separated most patients with APL. The AML samples presented the widest distribution in the PCA space, which was consistent with their heterogeneous nature and had some overlap with the ALL samples (Fig. 4a). The first two principal components (PC1, 53.1%; PC2, 30.9%) collectively explained 84.0% of the total variance in the dataset; however, complete class separation was not achieved (Fig. 4b, c). Next, we applied an ensemble of Isolation Forrest and Local Outlier Factor to reduce the proportion of distribution outliers and improve algorithm performance (Fig. 4d, e). Multicenter patients with confident predictions (cutoff > 0.9, $n = 2809$) served as a training set (Fig. 4d) for this pipeline and were tested on a hold-out dataset ($n = 2692$ with confidence < 0.9, Fig. 4e). Overall, this pipeline identified incorrectly predicted samples with twice as high a probability as correctly predicted samples (54.79% vs. 24.46%), improving the algorithm's accuracy while including as many patients as possible for

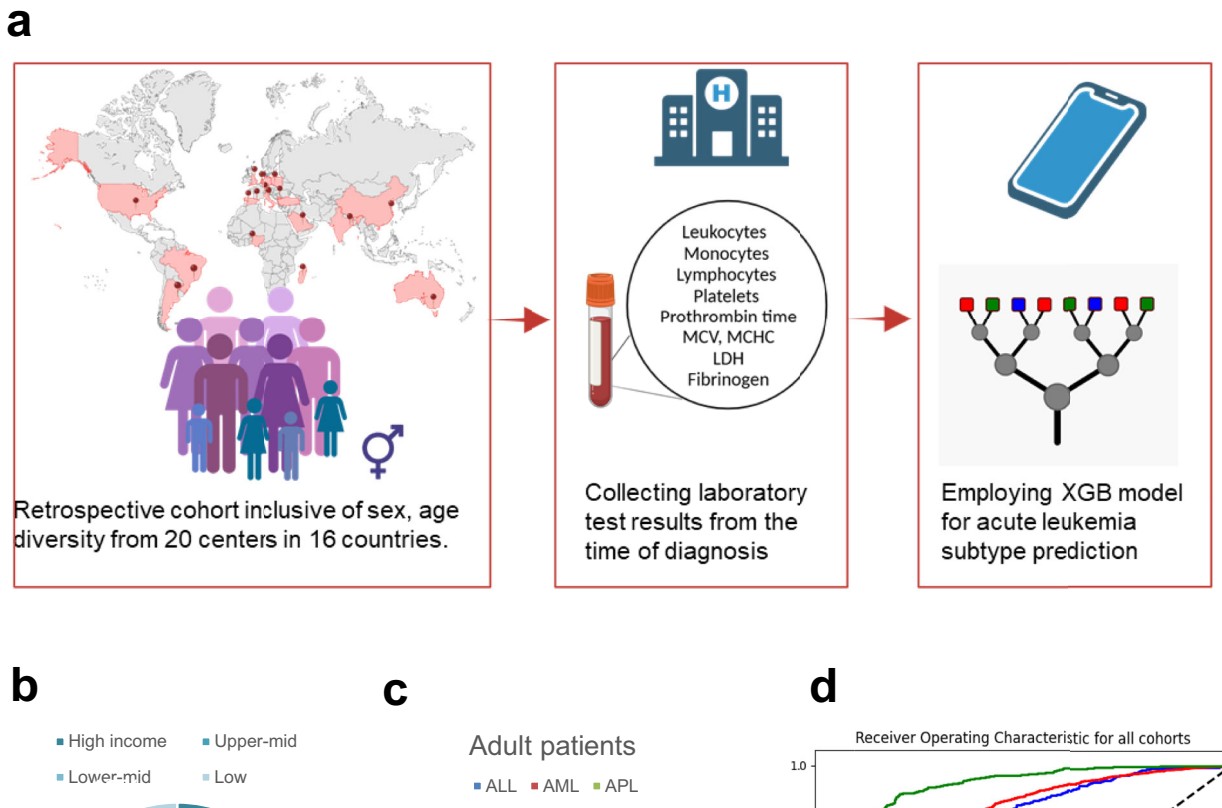

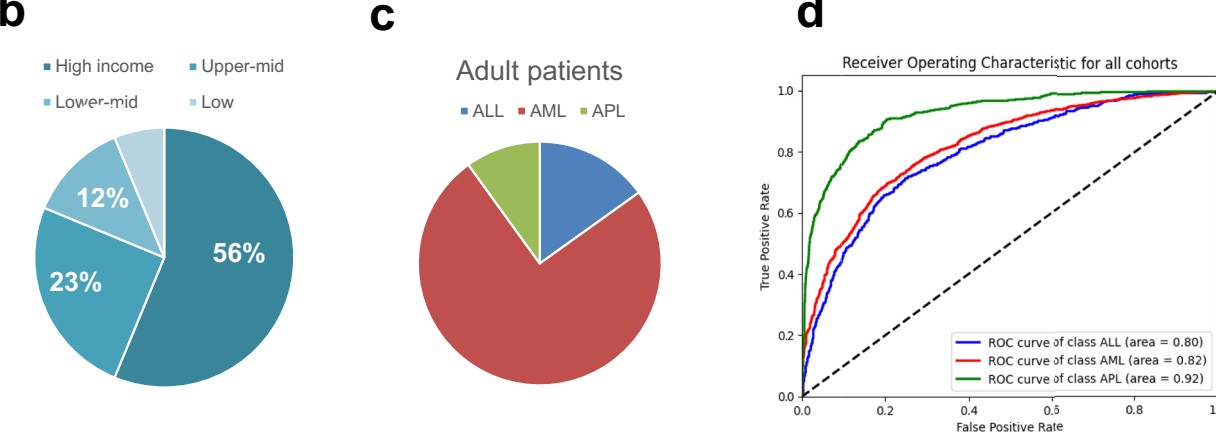

accurate predictions. Across diverse populations, the outlier detection approach consistently improved predictive performance (Fig. 4f, g). For AML, the accuracy improved in cohorts with high accuracy and linearly increased in cohorts with lower accuracy (Fig. 4h). The filtering preferentially removed incorrect predictions (AML 20.40% incorrect vs. 12.83% correct), and the isolated proportion was greater in the ALL class (65.98% vs. 27.56%). The sensitivity was increased for all classes,

especially for APL ( + 26.85%) and ALL ( + 18.54%). Stratified per center, the accuracy metrics revealed substantial improvements for most feature constellations (Supplementary Fig. 6). The developed pre-processing pipeline did also allow to challenge the algorithm with differential diagnoses to AML, ALL and APL presenting with leukocy-tosis or cytopenia (Supplementary Table 4). Cases with leukocytes above the normal were excluded with high accuracy (87.9%), while

**Fig. 1 | Building a global cohort for independent AI validation. a** Methodological study overview. Acute leukemia (AL) cohorts (*n* = 6,206 patients) were globally recruited, covering all continents. Twenty centers either shared anonymized datasets or performed federated testing using the provided python code. Laboratory data was extracted from electronic health records at the time point of leukemia diagnosis. Tabular or FHIR data entered the pipeline of an extreme Gradient Boosting (XGBoost) model. Given the number of variables and pipeline size, a mobile application on a cellphone or handheld device is feasible. Created with BioRender.com. **b** Pie chart of test cohort country distribution according to United Nations categorization and World Bank income category. Further information is detailed in Supplementary Table 2. **c** Pie chart of adult cohort composition: Acute myeloid leukemia (AML, red *n* = 3250), acute promyelocytic leukemia (APL, green, *n* = 432), acute lymphoblastic leukemia (ALL, blue, *n* = 655). **d** Accuracy of the model on the centrally tested adult cohort as measured by Area under the Receiver Operating Characteristic curves (AUROC), no cutoff applied (*n* = 4418, median age 55 years, 54.5% male.). Comparative line chart according to leukemia subtype: AML (red), APL (green) and ALL (blue). **e** Comparative accuracy by AUROC per disease subtype between confidence cutoff (*n* = 1061, median age 57 years, 58.7% male) dashed line and general algorithm. Left to right, AML (red), APL (green) and ALL (blue). Eligible patients per model are detailed in the figure legend.

cytopenic cases were more challenging (78.4% of leukopenia cases with other diagnoses excluded).

### Model retraining on pediatric data substantially improves precision

To further address health disparities in leukemia, we tested the algorithm's applicability among pediatric patients (*n* = 1746). ALL was the most common type of pediatric leukemia (Fig. 5a). Here, the pretrained model, originally developed on adult cohorts, achieved low accuracy metrics. The median AUROC for ALL was 0.75 (95% CI 0.74–0.75, Fig. 5b, c), far below the highest reported AUROC in adult ALL (Maastricht, The Netherlands, AUROC 0.92). Confidence cutoff predictions increased the accuracy but excluded too many patients (Fig. 5D). The ALL SHAP values were comparable to those of adults, but for AML, there were substantial differences (Supplementary Fig. 7a–c). As expected, pediatric patients with ALL had distinct feature ranges from centrally-tested adults, including significantly lower LDH levels (mean 558.46 U/l vs. 969.24 U/l); APL patients had higher PT percentages (mean 90.52% vs. 71.68% in adults) and higher monocyte counts (mean 9.41/nl vs. 2.88/nl in adults, Supplementary Data 3). Fibrinogen levels were lower than in adults, with particularly marked differences in AML patients (mean 2.63 g/l vs. 3.94 g/l in adults). Given the generalizability issues and unreliable precision-recall balance (Supplementary Data 4), we retrained the XGBoost on our pediatric cohort data resulting in substantially improved accuracy metrics with a 2000-fold bootstrap AUROC for ALL reaching 0.95 (95% CI 0.95–0.95, Fig. 5e). Pediatric APL was rare (*n* = 24), yet the pretrained predictions were accurate (e.g., AUROC Riyadh 0.90, Suzhou 0.97; Supplementary Fig. 7d). An age-decade stratified analysis of the entire cohort including adult and pediatric patients indicated that accuracy increased from childhood to adulthood and remains stable through older adulthood (Supplementary Fig. 8 and Supplementary Table 5). This result is encouraging for the algorithm's potential in APL detection in adults and children.

### Discussion

The number of AI studies in Hematology increases[20,21], however, few become tested on large international cohorts[21]. Hence, relevant issues regarding generalizability and robustness for potential real-world applications prevail. While the diagnosis of acute leukemia is guided by morphological examination of the peripheral blood and bone marrow smear, supplemented by flow cytometry and genetic testing[22], this study demonstrated on 6206 leukemia patients across 20 centers the global applicability of an AI algorithm to support rapid leukemia subtype detection from routine laboratory data. We also identified several issues and refined the AI pipeline with a multicentric distribution outlier detection approach to improve its robustness, accuracy and generalizability across leukemia subtypes and strengthen its practical usability.

Health disparities are a reality in cancer care[10,23], yet our aim should be to strive for health equity and address disparities by all available means. Among these in acute leukemia are rapid referral to specialized care centers and access to treatment. AI decision support may contribute to closing these gaps, as demonstrated by triage support in emergency departments[24]. However, in AI design, accessibility and affordability barriers should also be addressed to enable clinical implementation, particularly for patients in middle- and low-income countries. This algorithm built on the universal standardization of routine laboratory data (e.g., DIN ISO norms), which supports both robustness and generalizability but also accessibility and speed. Diagnosing acute leukemia according to the 2022 WHO[25] or ICC[26] criteria requires molecular and cytogenetic testing and flow cytometry, which are not readily accessible in most hospitals[14]. In high-income countries, these require express shipment with little delay, but many low- and middle- income countries have no central laboratory structure and overseas shipments either take time or are inaccessible[11], potentially reducing the accuracy of leukemia subtype diagnoses due to missing genetic or flow cytometric information. Bone marrow morphology is immediately accessible in cancer centers; however, expertise with microscopic examinations may vary[27] and require solid training. Hence, most AI studies in diagnostic hematology focused on AI-guided image recognition via digital-microscopy to support the morphological diagnosis of leukemia[17] and subtypes[28]. Others successfully leveraged deep learning for flow cytometry[29]. These convolutional neural networks excelled in accuracy, however, the implementation of licensed AI tools in expensive analyzers is today[30,31] and will likely continue in specialized laboratories of high-income countries[32] rather than becoming accessible in low- and middle-income countries. In such a setting, the gap between high-income settings prevails in the age of AI devices, meaning that disparities become perpetuated despite global advancements in UN development goals, which is an ethical AI issue[33]. In contrast, decision support by this algorithm is potentially universally implacable and may accelerate the diagnostic process and referral to specialized centers.

In their previous work, Alcazer et al. developed and tested an XGBoost algorithm on several French cohorts with high accuracy metrics using the confidence prediction cutoff[19]. However, this cutoff excluded 70-95% of our international validation cohort from predictions, reason why we focused on testing the algorithm without any cutoff, which showed high precision in some cohorts (e.g., Melbourne), but limited generalizability. How does this algorithm insert into the current landscape of emerging diagnostic approaches, including marrow morphology neural networks[17], Dirichlet mixture models[34] and AI supported rapid genomic sequencing?[35,36] It also leverages morphology patterns detectable from automatic leukocyte differentiation, cellular turnover (LDH) levels and acute phase molecules with coagulation parameters to differentiate AML from APL and ALL. Using routine laboratory data as AI features is promising because it is rapid and intuitive for physicians, supports both the robustness and generalizability of AI models and its inclusion previously substantially improved the accuracy of time-dependent predictions over static patient data[37]. Recent investigations have also explored non-reported features of CBC analyzers in hematologic conditions[38,39], which could further strengthen rapid point-of-care diagnostic models for leukemia. However, we observed variability in performance across international centers, some of which is accountable to the broad age range, sampling and distinct analytic devices, manufacturers or methodologies, which we also encountered in our study (e.g.,

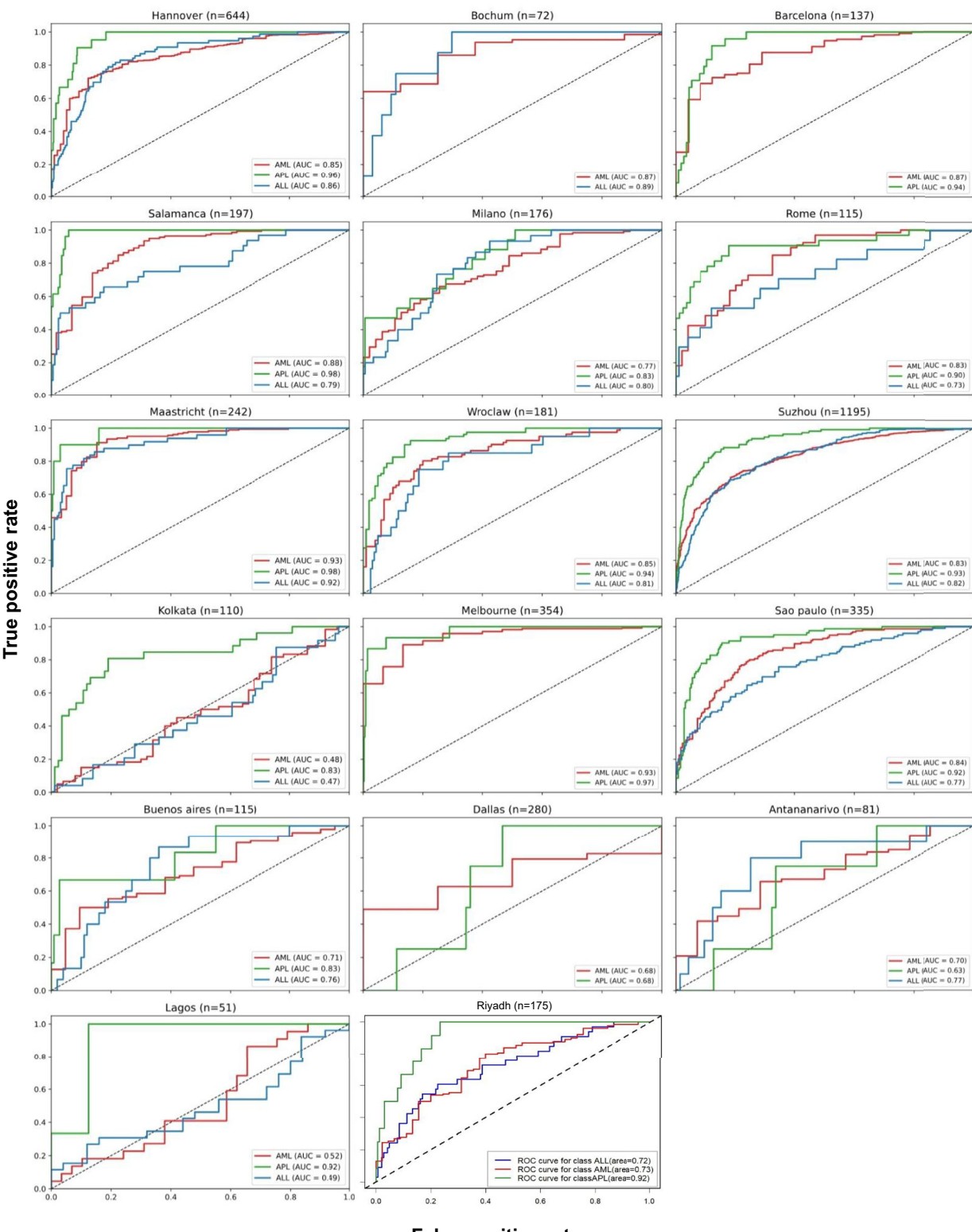

**Fig. 2 | Comparison of independent testing accuracy in adults according AU-ROC across sites before refinement.** Adult patients' AUROC metrics shown for each of the sites for AML (red), APL (green line) and ALL (blue). Europe: Hannover, Germany, Bochum, Germany, Barcelona, Spain, Salamanca, Spain, Milan, Italy, Rome, Italy, Maastricht, Netherlands, Wroclaw, Poland; Asia and Oceania: Suzhou, China, Kolkata, India, Melbourne, Australia, Riyadh, Saudi-Arabia; Americas and Africa: Sao Paulo, Brazil, Buenos Aires, Argentina, Dallas, TX, USA, Antananarivo, Madagascar, Lagos, Nigeria. Despite high accuracy metrics achieved in multiple centers across continents, the model's generalizability is impaired, with low accuracy observed in several sites.

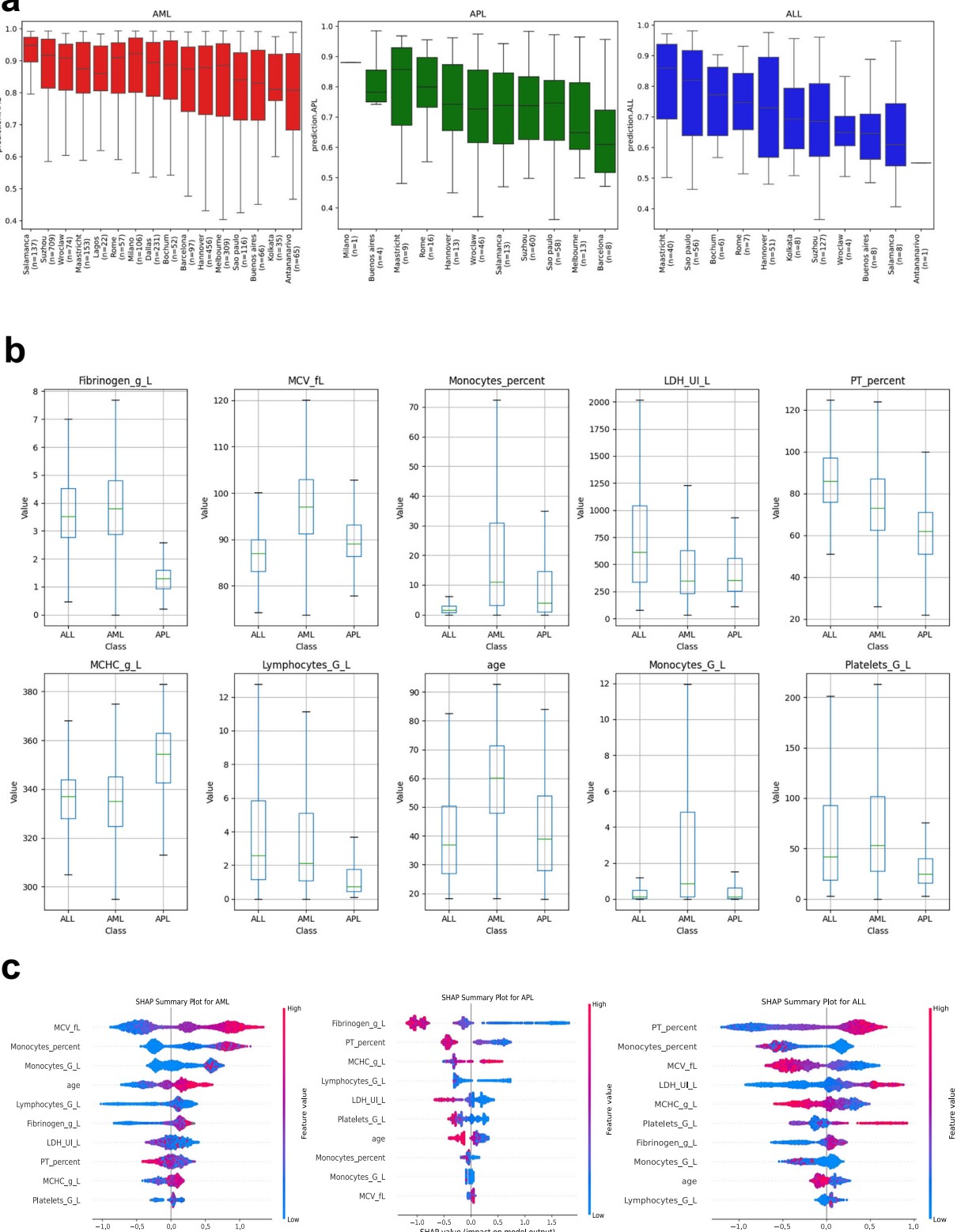

**Fig. 3 | Spectrum of true positive results across sites, feature distribution and feature importance. a** Adult patients' XGBoost true positive results ordered by acute leukemia subtype (left to right: AML (red), APL (green) and ALL (blue)) and site. The false negative results are shown in Supplementary Fig. 3b. Bars are given as median ± interquartile range. Whiskers correspond to 1.5 x interquartile range from box bounds. **b** Boxplot comparison of laboratory feature distributions and age between patients with true (positive and negative combined) predictions. Results are aggregated for each class, AML, APL and ALL. Bars are given as median ± interquartile range. Whiskers correspond to 1.5 x interquartile range from box bounds. **c** Model explainability by SHAPley values in the adult cohorts for APL, AML and ALL supports the importance of distinct features in each prediction setting. Features are ranked from top to bottom according to their importance (right side favors the subtype as e.g., AML). Each laboratory parameter is colored according to its value, with higher values being red.

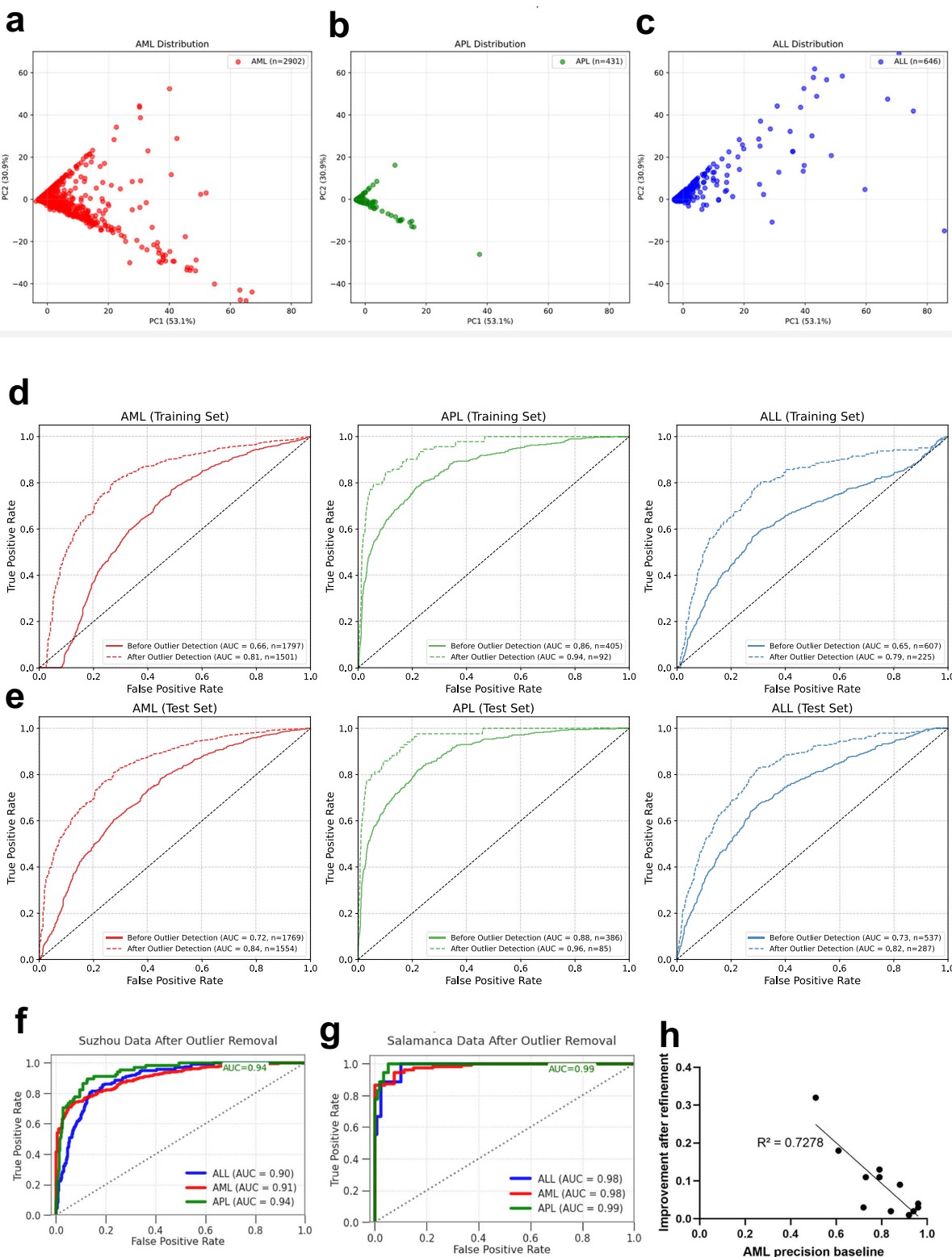

impedance sensors or fluorescence-optical sensors and their margins of accuracy), as well as systematically abnormal data of leukemia patients. The further away the data points were from the trained distributions, the worse they performed. Consequently, we developed a preprocessing pipeline and, importantly, a distribution outlier detection strategy, which was robust across different clinical settings and substantially improved model reliability without sacrificing generalizability.

Including a pediatric population for international testing was motivated by health disparities in children. The different age-dependent norms and leukemia phenotypes limited the pretrained algorithm performance in AML and ALL, reason why we retrained it on pediatric data to address this issue; nevertheless, its successful validation for pediatric patients with APL is encouraging. Interestingly, several hospitals in low- and lower-middle-income countries reported that coagulation tests are not routinely performed in leukemia patients

**Fig. 4 | Refining AI predictions for adults with leukemia. a–c** Dimensionality reduction by principal component analysis (PCA) of all feature's data used for the algorithm testing (as detailed in Supplementary Table 1). Adult cohorts by leukemia subtype (**a** AML, **b** APL and **c** ALL) reveals heterogeneity of AML with distinct feature patterns in APL and ALL, yet with overlap to AML. **d**, **e** Comparative accuracy by AUROC per disease subtype following distribution outlier detection via isolation forest (dashed line) versus general algorithm accuracy. Left to right, AML (red), APL (green) and ALL (blue). Eligible patients per model are detailed in the figure legend. **d** Isolation Forest outlier detection training cohort. Training of outlier detection pipeline on patients with predictions > 0.9 confidence cutoff and their normalized performance. **e** Isolation Forest outlier detection test cohort. Test of improved performance in the hold-out dataset of patients previously not-reaching the 0.9 confidence cutoff via outlier detection with isolation forest. The degree of improvement is robust for all classes, both in development and test cohorts. **f, g** Exemplary differences in single-center AUROC after outlier detection (**f** Suzhou, China, left and **g** Salamanca, Spain, right) across diverse geographic and socio-economic settings (e.g., high income vs. upper-middle income country, province vs. metropolis, Europe vs. East-Asia). **h** Linear regression analysis. Dot plot correlation of AML precision metrics before improvement (x-axis) and absolute improvement in precision (y-axis), all centrally tested adult AML patients, each center representing one dot. The improvement is linearly increasing for centers with lower initial precision metrics.

because of costs. As coagulation at diagnosis is among the most important features of this leukemia subtype classifier and is not very costly, our results support that coagulation at diagnosis must be a global standard to support the early identification of APL.

In conclusion, this international study testing an AI model to predict acute leukemia subtypes provides important results that support the promise of inclusive AI tools to reduce access barriers in hematology and support health equity. While the gold standard for leukemia diagnostics remains unaffected, such an algorithm may accelerate referral to specialized centers or, in resource-poor settings, eventually complement standard diagnostics. This study has strengths, including its large sample size, diverse cohort and model refinement pipeline, and limitations, such as the imbalance of total patient numbers from lower-middle- and low-income countries, which have been difficult to recruit, and many contacted centers have not determined coagulation parameters and LDH levels routinely at diagnosis. The distinct incidence of each leukemia subtype resulted in model class imbalance. Anticoagulation with cumarin derivatives may interfere with model performance, as few AML patients with impaired coagulation (e.g., PT < 60) and normal leukocytes were misclassified as APL. Leukemic presentation associated with higher accuracy metrics than cytopenia, also for excluding differential diagnoses. This algorithm does not provide detailed genetic AML subclasses but identifies APL or AML with clinically meaningful accuracy, thus supporting physicians in their initial decisions and care. Prospective trials are warranted to test whether an accelerated leukemia diagnosis via an AI-supported medical device can lower early mortality rates in low- and middle-income countries, yet this independent testing and model refinement paved the way for such trials.

## Methods

### Statement of ethics
This study was conducted according to the Declaration of Helsinki and approved by the Ethics Committee of the Medical Faculty of the University Duisburg-Essen, approval number 24-11882-BO, as well as by further local IRBs (Dallas, Hannover, Maastricht, Riyadh, Suzhou). Given the use of anonymized routine laboratory data as features, informed consent was waived by the IRB. No participant compensation was provided. This study has been registered in the German Registry for Clinical Trials (DRKS). https://drks.de/search/en/trial/DRKS00037360.

### Study cohort and data
To account for data diversity during independent testing and model refinement, we retrospectively assembled international patient cohorts with acute leukemia, whose laboratory features included the complete blood cell (CBC) count (total white blood cell count (WBC; g/L or $10^9$ per L), monocyte and lymphocyte counts (g/L or $10^9$ per L), platelet count (g/L or $10^9$ per L), mean corpuscular volume (MCV; fL), mean corpuscular hemoglobin concentration (MCHC; g/L), lactate dehydrogenase (LDH; IU/L), fibrinogen (g/L) and prothrombin time (%) at the earliest possible point in leukemia diagnosis (i.e., at hospital admission). Data on prothrombin time reported in seconds were

converted into percentages via calibration plasma. In addition, meta-data such as age, sex, and, if available, European Leukemia Net (ELN) 2022 risk[4] and World Health Organization (WHO) 2022 classification[25] information were included and considered for subset analyses. Sex was self-reported. The features are detailed in Supplementary Table 1. Pediatric (< 18 years) and adult patients (≥ 18 years) were analyzed separately. Patients were included from high-, upper middle-, lower middle and low- income countries according to the World Bank and United Nations (UN) classification[40] and categorized using the UN Human Development Index[41] (accessed February 2025, Supplementary Table 2). Most countries have combined state and private health systems for cancer care[42,43].

### Data preparation
The methodology overview is illustrated in Supplementary Fig. 1. This study was conducted primarily in a centralized manner with transfer of anonymized data (n = 19). In addition, the site in Riyadh, Saudi-Arabia, implemented and tested the algorithm locally. Cohorts were provided by contributing investigators either through a custom Fast Healthcare Interoperability Resources (FHIR) pipeline or as file data imports (e.g.,.csv files). For FHIR-derived data, the relevant FHIR resources were merged to construct feature-label datasets for predictions. To ensure consistency across data sources, some features of centrally submitted laboratory data were transformed into SI units, which were required to run the algorithm via a configuration file that facilitated the reproducible processing of new cohorts. A laboratory medicine expert manually inspected all centrally collected data for potential errors and excluded technically implausible samples (n = 10). The multicenter data underwent programmatic preprocessing, which included filtering for quality and removing samples with more than 20% missing features (n = 439). These processes are visualized in a STROBE[44] diagram (Supplementary Fig. 2). In the cohort from Antananarivo, Madagascar, coagulation markers (PT and fibrinogen) and LDH were systematically unavailable; exceptionally, these patient data were tested despite the greater proportion of missing data.

### Model evaluation
For independent, international testing, we conducted several experiments. First, acute leukemia subtype predictions were generated by executing the pretrained eXtreme Gradient Boosting (XGBoost) model[19] to the prepared datasets. The performance of the AI classifier was evaluated via various metrics, including the area under the receiver operating characteristic curve (AUROC), sensitivity, specificity, F1-score, and confusion matrices. Bootstrapping (2000-fold) was employed to estimate metric means and generate 95% confidence intervals (CIs). Bootstrapping required a minimum cohort size of 30 valid samples per center. The analysis was stratified by leukemia subtype to allow for granular performance evaluation. Subclasses containing fewer than ten samples were not reported to maintain statistical validity. We analyzed the performance separately for each center and adult and pediatric patients to account for data variance. The XGBoost algorithm provides information on the confidence of predictions, which may be leveraged to define clinically

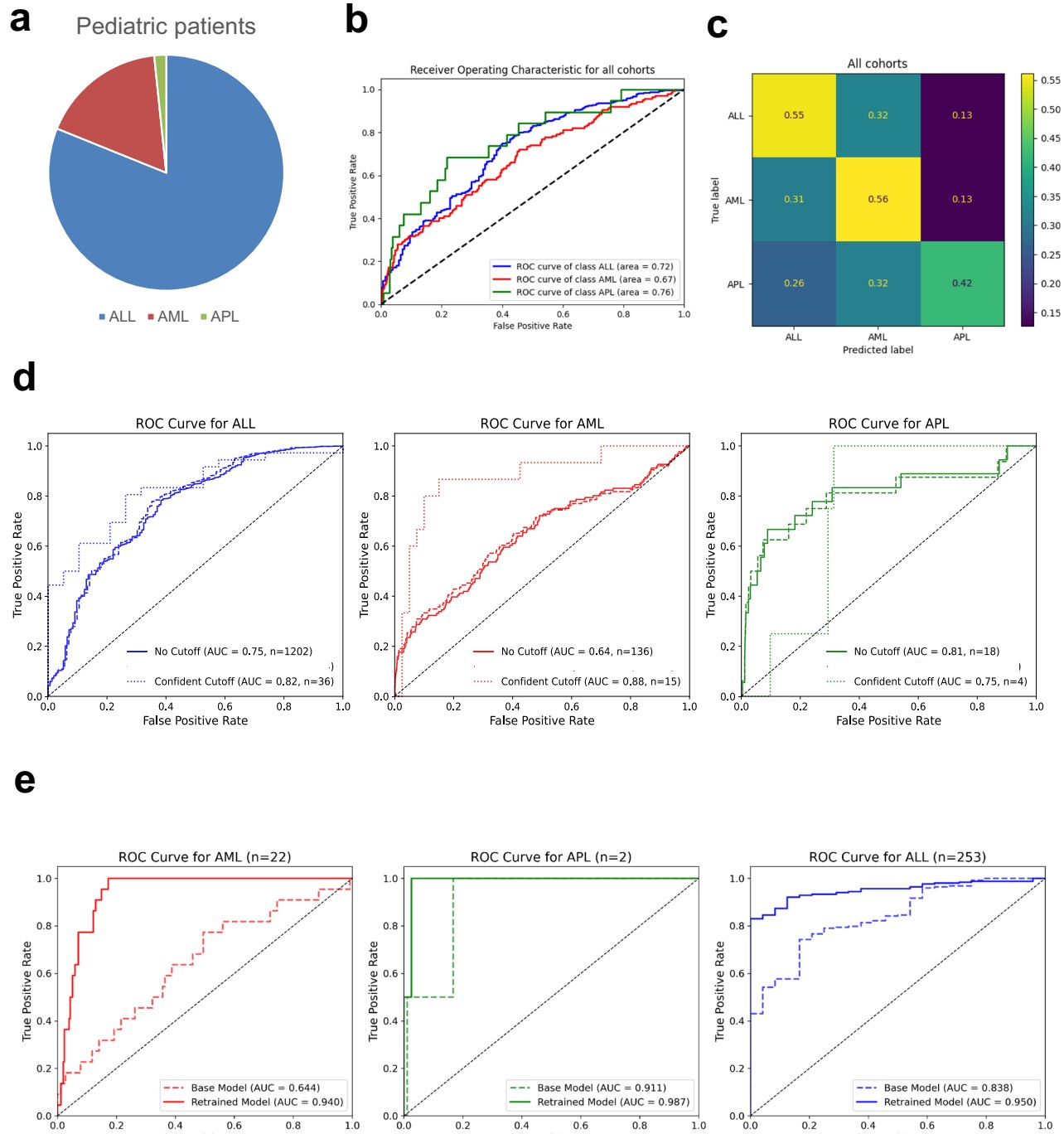

**Fig. 5 | Validation of the APL predictor in the pediatric cohort. Lower accuracy for AML. a** Pie chart of pediatric cohort composition: Acute myeloid leukemia (AML, red), acute promyelocytic leukemia (APL, green), acute lymphoblastic leukemia (ALL, blue). **b** Accuracy of the pretrained model on the centrally tested pediatric patients as measured by Area under the Receiver Operating Characteristic curves (AUROC), no cutoff applied (*n* = 1414, median age 6 years, 48.5% male). Comparative line chart according to leukemia subtype: AML (red), APL (green) and ALL (blue). **c** Confusion matrix of the centrally tested pediatric cohort. Results in proportions. **d** Comparative accuracy by AUROC per disease subtype between the confidence cutoff dashed line and the general algorithm. Left to right, AML (red), APL (green) and ALL (blue). Eligible patients per model are detailed in the figure legend. **e** Accuracy of the model retrained on pediatric patients as measured by Area under the Receiver Operating Characteristic curves (AUROC); multicenter test set no cutoff applied (*n* = 277). As a comparator baseline pretrained model on the set (dotted line) and retrained model on the set (line). Comparative line chart according to leukemia subtype: AML (red), APL (green) and ALL (blue).

meaningful cutoff values for potential clinical implementation. For direct comparison of our results to those obtained by Alcazer et al., we next applied their confidence cutoff strategy using positive predictive value (PPV) thresholds (AML: 0.945, APL: 0.749, ALL: 0.959) as described previously[19]. We also tested the applicability in specific clinically relevant subgroups (e.g., ELN-defined). Third, we compared the feature importance via SHapley Additive exPlanations (SHAP) values. The feature distributions of correctly classified and misclassified patients were comparatively analyzed for each predicted class using boxplots. We also employed principal component analysis (PCA) on the features to check whether the classes were linearly separable.

## Model refinement

Given that pediatric patients have distinct laboratory ranges and feature constellations, which the XGBoost algorithm was not previously exposed to, we retrained it on our pediatric dataset, split into training (80%) and test (20%) sets. The retraining was performed in R using stratified sampling to preserve the class distribution. Performance was evaluated on the hold-out test set as described above. We chose not to retrain the model on adult patient data because we aimed preparing the algorithm for clinical testing, robust to future unknown feature constellations. The subsequent steps aimed to standardize the applicability of the pretrained model, to clarify when it offers trustable results, and to enhance both robustness and generalizability. The distribution outlier detection pipeline combined Isolation Forest and Local Outlier Factor models trained per class on preprocessed, normalized data obtained from adult patients (Supplementary Fig. 1). Centers contributing more than 30 samples were included to maintain statistical reliability. The dataset was partitioned into two sets: a high-confidence training set comprising samples with prediction confidence of 0.9 or higher and a lower-confidence test set. Before Isolation Forest training, the data underwent a series of preprocessing steps; this included imputing missing values via the median method and normalizing data via standard scaling, to achieve a zero mean and a unit variance. The training phase employed a class-specific approach, where separate models were trained for each disease class. Each class incorporated an Isolation Forest with automatic contamination detection and a Local Outlier Factor model configured with 20 neighbors and automatic contamination settings. The test data underwent these same preprocessing steps. The performance was evaluated using AUROC scores and classification reports, which compared the results before and after removing distribution outliers.

## Software and implementation

The original algorithms had previously been implemented in R and made publicly available (https://github.com/VincentAlcazer/AIPAL). We selected the best performer on the development cohort, an extreme gradient boosting (XGBoost) algorithm, as the pre-trained model for this study (The workflow is illustrated in Supplementary Fig. 1). For independent testing, we developed a custom FHIR pipeline using Python (version ≥ 3.10) leveraging multiple open-source packages: FHIR-PYrate for FHIR data handling, Weights & Biases (wandb) for experiment tracking, SQLAlchemy and psycopg2-binary for database operations, PyYAML and python-dotenv for configuration management, and various data science libraries including matplotlib, scikit-learn, XGBoost, SHAP, and UMAP for analysis and visualization. Details and references to all libraries are provided in the Supplementary Methods.

Further, we created a standardized prediction workflow pipeline that can ensure reproducibility of methods across all cohorts and to ensure the validity of the predictions for a broader patient population. It included pre-processing the data, making predictions and evaluating the results. This validation pipeline employed a combination of the R programming language and Python. From electronic medical records, the pipeline identified the first occurrence of coded leukemia diagnoses and extracted the required laboratory results at diagnosis, as well as age and sex. The same Python pipeline was employed for importing data for leukemia subtype detection. This comprehensive and reproducible pipeline ensured the robust validation of the leukemia subtype classifier across diverse data sources and settings. Further, the pipeline was extended to allow new samples to be validated by the developed outlier detection algorithm. Probabilistic predictions for leukemia subtypes (AML, APL, ALL) and other evaluation results were logged to Weights & Biases (https://wandb.ai/), including cohort characteristics, performance metrics, and outlier statistics. Results were also tabulated and stored in .CSV files for further analysis. The code of the validation Python pipeline, including the outlier detection

algorithm, is publicly available in the UMEssen GitHub repository (https://github.com/UMEssen/aipal-validation).

## Inclusion and ethics statement

All local contributors to this research were included as authors or acknowledged when not fulfilling authorship criteria. We confirm that this research has local, regional and global impact and has been designed and conducted in collaboration with local communities.

## Reporting summary

Further information on research design is available in the Nature Portfolio Reporting Summary linked to this article.

# Data availability

Anonymized acute leukemia laboratory data is deposited to the HARMONY Alliance data repository due to IRB restrictions. Data access can be requested by researchers in written form via office@harmony-alliance.eu (expected response time 2 weeks) and is subject to HARMONY data access committee approval. HARMONY will make data available for up to 12 months. Source data are provided with this paper. A demo dataset is provided with the GitHub code. Source data are provided in this paper.

# Code availability

The pipeline code is publicly available (https://github.com/UMEssen/aipal-validation) under the MIT License. The version of the code used in this study is citable via Engelke M and Turki AT, International testing and refinement of an AI algorithm to predict acute leukemia subtypes from routine laboratory data, 2026[45]. https://doi.org/10.5281/zenodo.18436461.

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

## Acknowledgements

This work was partially presented as an oral presentation at the 66th Annual Meeting of the American Society of Hematology (ASH) 2024 in San Diego: Turki AT et al. Independent, International Validation and Refinement of a Machine Learning Algorithm to Classify Acute Leukemia Using Routine Laboratory Features, Blood (2024) 144 (Suppl 1): 790. https://doi.org/10.1182/blood-2024-207058). Another abstract of this research was presented as an oral presentation at the 30th Congress of the European Hematology Association (EHA) 2025 in Milan. https://library.ehaweb.org/eha/2025/eha2025-congress/4159411/merlin.engelke.refinement.and.international.validation.of.a.machine.learning.html?f=menu%3D6%2Abrowseby%3D8%2Asortby%3D2%2Amedia%3D3%2Ace_id%3D2882%2Aot_id%3D31588%2Amarker%3D5843%2Afeatured%3D19588. We would like to thank Dr. Mehmet Fatih Orhan, Dr. Berkin Berk, Dr. Yilmaz Ay, Dr. Alessandro Buizza and Dr. María Ines Paganini for supporting data preparation to this study, Prof. Andrew Wei, Dr. Amir Toor, Dr. Titilope Adeyemo, Dr. Karina Tozatto-Maio, Dr. Valentín Ortiz-Maldonado, Dr. Christian Fanomezana, Prof. Dirk Reinhardt and Prof. Uta Dirksen for their support building the consortium, Prof. Shadi Albaroquni and Katja Scheidler for critical advice and review.

## Author contributions

A.T.T. designed the study, assembled the cohort, supervised the model refinement, analyzed and interpreted data and wrote the manuscript. Y.F., A.H.S., W.S., S.F., K.Y., C.V.E., Y.M., M.K., M.G.D.P., A.M.R., L.G., E.A., M.M.R., N.R.N., D.K.M., A.T.B., S.N.M., A.A., A.E., R.M., M.T.V., J.C., A.M., M.B.V., A.K., P.M., D.S., R.S.F., D.M., G.A., M.M., M.K., E.R., F.T., M.S., and D.W. contributed anonymous cohort data for analysis and contributed to the data interpretation. M.A. and T.A.E. performed localized testing leveraging our Python pipeline for federated testing. K.S. provided laboratory medicine expertise and manually reviewed data. F.N. acquired funding and contributed to the interpretation. J.K. revised the manuscript and provided advice. M.E. developed the FHIR-compatible Python pipeline, performed model validation, developed the model refinement pipeline, analyzed and interpreted data. L.H. contributed to the validation pipeline and analyzed data. All authors approved the final manuscript.

## Funding

## Competing interests

A.T.T.: Consultancy for Maat Pharma, Onkowissen.tv, CSL Behring, Biomarin, Oncopeptides and Pfizer, research funding by Neovii, travel reimbursement by Novartis and Neovii; YFM has received honoraria/consulting fees from BMS, Kura Oncology, BluePrint Medicines, Geron, OncLive and MD Education, VJHemOnc, Curio Science, and Medscape Live. YFM participated in advisory boards and received honoraria from Stemline Therapeutics, Blueprint Medicines, Taiho Oncology, SOBI, Rigel Pharmaceuticals, Geron, Cogent Biosciences and Abbvie. YFM received travel reimbursement from MD Education; A.M.R.: Consultant or advisory role (Bristol Myers Squibb, Abbvie, Kite Gilead), travel grants (Kite Gilead, Roche, Takeda, Janssen, Abbvie, Jazz Pharmaceuticals), speaker (Abbvie, Gilead). M.T.V: Consultant or advisory role (Bristol Myers Squibb, Novartis, Sobi, AOP), travel grants (Novartis), speaker (Bristol Myers Squibb, Novartis, Sobi, AOP); M.S.: Consultant or advisory role (Bristol Myers Squibb, Astellas Servier), travel grants (Servier), speaker (Bristol Myers Squibb, Astellas, Servier, Abbvie, Daychi-Sankyo). The remaining authors declare no competing interests.

## Additional information

[1]Computational Hematology Lab, Institute for Artificial Intelligence in Medicine, University Hospital Essen, Essen, Germany. [2]Department of Hematology and Oncology, University Hospital Marienhospital, Ruhr-University Bochum, Bochum, Germany. [3]Institute for Artificial Intelligence in Medicine, University Hospital Essen, Essen, Germany. [4]The First Affiliated Hospital of Soochow University, Suzhou, China. [5]Department of Hematology, University Hospital of Salamanca, Salamanca, Spain. [6]Hospital das Clinicas da Faculdade de Medicina da Universidade de Sao Paulo, Sao Paulo, Brazil. [7]The Alfred Hospital, Melbourne, Australia. [8]Bahcesehir University Medicalpark Goztepe Hospital, Department of Pediatric Hematology/Oncology, Istanbul, Turkey. [9]Maastricht University Medical Center, Department of Internal Medicine, Division of Hematology & GROW research, Institute of Oncology and Reproduction, Maastricht, The Netherlands. [10]University of Texas Southwestern, Dallas, USA. [11]Department of Hematology, Cell Therapies and Internal Medicine, Wroclaw Medical University, Wrocław, Poland. [12]Department of Hematology, University Hospital N° 2 in Bydgoszcz, Collegium Medicum in Bydgoszcz, Nicolaus Copernicus University in Toruń, Bydgoszcz, Poland. [13]King Faisal Specialist Hospital & Research Center, Riyadh, Saudi Arabia. [14]IRCCS Humanitas Research Hospital & Humanitas University Milan, Milan, Italy. [15]Department of Hematology, Hospital Clínic Barcelona, Barcelona, Spain. [16]Tor Vergata University Roma, Rome, Italy. [17]Central Laboratory Division, University Hospital Marienhospital, Ruhr-University Bochum, Bochum, Germany. [18]Department of Pediatrics III, University Hospital Essen, Essen, Germany. [19]Hospital Universitario Austral, Buenos-Aires, Argentina. [20]Tata Medical Center, Kolkata, India. [21]Hematology Department, University Hospital Joseph Ravoahangy Andrianavalona of Antananarivo, Antananarivo, Madagascar. [22]Lagos University, Lagos, Nigeria. [23]Leukaemia Research Cytogenomics Group, Translational and Clinical Research Institute, Newcastle University, Newcastle upon Tyne, United Kingdom. [24]Department of Hematology, Hannover Medical School, Hannover, Germany. [25]Alfaisal University, Riyadh, Saudi Arabia. [26]Marmara University Faculty of Medicine, Department of Pediatric Hematology and Oncology, Istanbul, Turkey. ✉e-mail: amin.turki@uk-essen.de

