## [Peer Review File · Nature Communications]

International testing and refinement of AI algorithms predicting acute leukemia subtypes from routine laboratory data

Corresponding Author: Dr Amin Turki

Version 0:

Reviewer comments:

Reviewer #1

(Remarks to the Author)

The study by Turki describes a AI algorithm for predicting acute leukemia subtypes (ALL, APL and AML) from routine laboratory data, such as LDH, PT, age, MCHC, fibrinogen, MCV, monocytes, lymphocytes and platelets. Impressively they test this on over 6000 cases from 20 worldwide centers across the age spectrum. They first tested a previously established XGBoost model across the cohort and within specific centers and then refined the model for better performance. Overall, this is a timely and interesting study. A few concerns need to be addressed prior to publication:

1. The variability in performance across the different centers is significant yet there isn't enough discussion regarding the source of the variability.
2. The monocyte count appears to lead to issues with predictions. Does removal of monocytes from the prediction have a positive or negative impact?
3. Although it makes sense to have a separate predictor for kids and adults, the firm cutoff of 18 could lead to issues. The team should show the performance of each classifier across the age spectrum (perhaps broken down into decades). It is possible that the a strict cutoff is not ideal.
4. The discussion is the same length as the results and is too long. Please shorten to focus on the take away messages rather than summarizing the results.

(Remarks on code availability)

Reviewer #2

(Remarks to the Author)

Turki et al. have compiled an international multicentric dataset of routine laboratory parameters from a diverse group of over 6000 patients and used it to improve an approach introduced previously by Alcazer et al. in 2024 on a French cohort and develop a group of algorithms for subtyping acute leukemia. The topic responds to a relevant medical question, namely quickly and easily subtyping AML patients, which has implications on treatment and potentially survival of this patient population. They also address subtyping in pediatric populations. Overall, the paper is well written. While the machine learning approach is largely derived from previous work, the machine learning approach used is well documented and the code made available.

Apart from these strengths, there are a few points that the authors should address before the manuscript is ready for publication.

Major points:

- 1) Large part of the scientific value of this work justifying publication in a journal like Nature Communications lies in compiling a unique and diverse global dataset of leukemia patients, especially as the algorithmic strategy has been described

previously. Hence, data accessibility to the scientific community is key when considering the scientific impact of the manuscript. The authors address this point by intending to publicly share the dataset via the HARMONY Alliance repository. However, they are very vague on the amount of data or centers included in the final deposit ("centers willing to share their data"). For high relevance in the community, the anonymized dataset should be deposited in full or with minimal redactions in order to be accessible for independent testing and as a basis for future work. The authors should clearly state what they are willing to share in a revised version of the manuscript.

2) The authors' outlier filtering method is at the heart of their refinement, but its description is unclear. In the relevant section, the authors state "Overall, this pipeline identified incorrectly predicted samples with twice as high a probability as correctly predicted samples (54.79% vs. 24.46%), improving the algorithm's accuracy while including as many patients as possible for accurate predictions." It is not clear what the prediction then is for these cases. Are they excluded from the analysis? Also, could there be any pattern shared by the outliers removed that is not captured by the algorithm (e.g. secondary diagnoses or lab errors)?

3) Before model refinement, there is considerable inter-site difference for the dataset, down to models that do not work at all (AUROC 0.5) in some cases, as shown in Fig. 2. Can the authors provide this full analysis also for the post-refinement model? This would provide a more comprehensive overview of the site-specific impact of their refinement method.

4) While the authors achieve a substantial improvement using their outlier removal strategy, classification performance based on lab parameters is still far from the diagnostic gold standard. What strategies to the authors foresee to improve from this point? Do they believe that further increasing case numbers or including other clinical values would be important?

Minor points:

1) When considering pediatric cases, the authors find that different relevant lab value constellations differ from the adult case. Do the authors believe that this reflects the distinct biology of pediatric disease, or is it due to the different case number distribution

2) Presentation: Figures are overall quite busy, with small legends, especially in Fig. 2 and 3.

3) In Fig.2, the same formatting should be used for all graphs, and multiple similar legends avoided.

(Remarks on code availability)

Reviewer #3

(Remarks to the Author)

In this manuscript, the authors have validated and improved upon a published model for the prediction of acute leukemia subtype across several centres. They use CBC counts, blood LDH levels, and coagulation parameters to predict the leukemia subtypes. The concept, overall, is promising and tries to solve an important problem, i.e., lack of access to sophisticated investigations in developing countries. That being appreciated, these are the concerns that are noted:

1. Patients with acute leukemia (AL) can have varied clinical presentations from frankly leukemic (elevated WBC counts) to cytopenic. Frank leukemic presentations are also noted in other myeloid malignancies such as chronic myeloid leukemia and other MPNs, as well as MDS-MPN overlap conditions, such as chronic myelomonocytic leukemia in adults and juvenile myelomonocytic leukemia in children, which have monocytosis. These presentations are also seen in lymphomas such as chronic lymphocytic leukemia or mantle cell lymphoma, to name a few. Similarly, AL also commonly presents with frank cytopenia mimicking other malignancies, for example, MDS, CMML (dysplastic type), and reactive conditions (eg, B12/Folate deficiency, sepsis, or chemotherapy-induced cytopenia). The training set that is used for the model uses features of frankly leukemic patients and, in all probability, excludes patients with other conditions or leukemia that have been partly treated with disease-modifying therapy (as stated in their Lancet Digital Health paper). This is a major shortcoming of the study, as these are common problems in the real world and are amplified in LMIC settings, which is the premise of this paper.

2. Validation studies to refute the diagnosis of acute leukemia: This ML model should be validated on conditions that are not acute leukemia but present with leucocytosis and monocytosis (lymphoma/chronic lymphoproliferative disorders, MDS-MPN overlap syndromes), as well as cytopenia such as MDS, MDS-MPN, and acute leukemia with cytopenic counts. This will ensure that this model has true clinical value and is applicable to a resource constrained setting.

3. It is unclear how patients were selected for training the model or for validation. Are these consecutively diagnosed patients, or is there a selection bias?

4. There are minor errors in the supplementary data as total WBC, monocyte, platelet counts are in G/L

(Remarks on code availability)

Yes, the github repository does have info on how to run the pipeline and generate synthetic data
<https://github.com/UMEssen/aipal-validation>

Version 1:

Reviewer comments:

Reviewer #1

(Remarks to the Author)

The authors have satisfactorily addressed my comments.

(Remarks on code availability)

Reviewer #3

(Remarks to the Author)

The authors have answered all queries. no further comments

(Remarks on code availability)

The authors have answered all queries. no further comments

Reviewer #4

(Remarks to the Author)

The authors have adequately addressed all the points raised.

However, several aspects of the abstract remain unclear and would benefit from clarification:

L89: The authors first report the performance of the “confident” AI-PAL predictions (AUROC 0.94/0.98/0.84 for acute myeloid leukemia/acute promyelocytic leukemia/acute lymphoblastic leukemia). Are these results derived from the overall merged dataset, or from a specific cohort or data split (e.g., test set)?

L93: The AUROC reported for acute myeloid leukemia is 0.72 here, whereas 0.94 is reported immediately before. Since the test set is not defined in the abstract, it is difficult to interpret what this comparison refers to (e.g., random split of the pooled cohort, external dataset, or a selected subset).

I therefore suggest a minor revision of the abstract to clearly specify what is being referred to in terms of the baseline model and the evaluated cohorts.

(Remarks on code availability)

made.

Point by point response to peer review for

"International testing and refinement of an AI algorithm to predict acute leukemia subtypes from routine laboratory data" by Turki et al.

Reviewer 1 report:

(Remarks to the Author):

The study by Turki describes a AI algorithm for predicting acute leukemia subtypes (ALL, APL, and AML) from routine laboratory data, such as LDH, PT, age, MCHC, fibrinogen, MCV, monocytes, lymphocytes, and platelets. Impressively they test this on over 6000 cases from 20 worldwide centers across the age spectrum. They first tested a previously established XGBoost model across the cohort and within specific centers and then refined the model for better performance. Overall, this is a timely and interesting study. A few concerns need to be addressed prior to publication:

Response: Thank you very much for the positive evaluation of our work.

1. The variability in performance across the different centers is significant yet there isn't enough discussion regarding the source of the variability.

Response: We have extended the discussion on page 13, lines 352-358, to include possible sources of variability between centers. This now includes population characteristics and feature distributions such as age, pre-analytics, and - to a minor extent - laboratory analyzers. Our analysis indicated that the further away the collective data points (e.g. leukocytes) were from the laboratory results of the trained distributions, the worse they performed. This explains also the lower performance in leukopenia than in leukocytosis. The systematically abnormal results in leukemia patients remain a challenge to defining a robust pattern, which, however, was addressed by the preprocessing pipeline.

2. The monocyte count appears to lead to issues with predictions. Does removal of monocytes from the prediction have a positive or negative impact?

Response: Following this suggestion, we tested an additional model without monocytes, which did not improve the performance in adults but increased F1 scores in pediatric AML patients. The following performance metrics were recorded:

Adult patients (n=3,173, monocyte excluded model)

- ALL: AUC 0.79 (unchanged), F1 0.48 (+0.01)

- AML: AUC 0.83 (+0.01), F1 0.83 (-0.02)

- APL: AUC 0.92 (unchanged), F1 0.65 (+0.05)

Pediatric patients (n=517, monocyte excluded model)

- ALL: AUC 0.69 (-0.05), F1 0.74 (unchanged)

- AML: AUC 0.62 (-0.02), F1 0.40 (+0.15)

- APL: AUC 0.77 (-0.03), F1 0.16 (+0.01)

However, excluding monocyte variables (both absolute counts and %) forced us to remove many patients with missing features, as we only allowed up to 20% missing features. This removal reduced the adult cohort by 25% (to n=3,173, monocyte excluded model) and the pediatric cohort by 66% (to n=517, monocyte excluded model). The AUC metrics remained largely unchanged for adults while the F1 improved for pediatric AML. We think that the altered dataset composition has impacted the performance metrics by selecting those datasets without or with less missing features. Improvements in pediatric AML may be the consequence of improved data quality in the subset rather than the true effect of removing monocyte features. Given that substantial changes were only observed for pediatric AML and to preserve sample size, generalizability, and robust population representation, we decided to keep the monocyte features for the final model.

3. Although it makes sense to have a separate predictor for kids and adults, the firm cutoff of 18 could lead to issues. The team should show the performance of each classifier across the age spectrum (perhaps broken down into decades). It is possible that the a strict cutoff is not ideal.

Response: Indeed, the biological separation between adult and childhood leukemia is not strictly drawn at the age of 18, however, there were several reasons why we adopted it. First, 18 years is a common transition age from pediatric to adult care. Second, Alcazer et al. *Lancet digit. Health* 2024 trained their algorithm only on patients aged 18 and older, and one of the objectives of our study was to validate that algorithm. Hence, it appeared reasonable to adopt the 18-year cutoff.

Yet, we followed the suggestion of Reviewer 1 and comparatively tested the entire cohort, including both adult and pediatric patients, with age cutoffs at each decade on the model without confidence cutoffs. The AUC metrics indicate that accuracy increases gradually from childhood to young adulthood and remains stable through older adulthood (particularly for AML and APL). The best AUCs were observed for middle-aged patients around the median age of the patient population in our study. Importantly, the APL prediction remains at high accuracy (AUC 0,94) until the age cutoff of 80 years. Two specific subsets were not fully evaluable due to low case numbers (APL in patients aged 0-9 years (n=2); ALL in patients aged 80-89 years

(n=6)). We added this decade-stratified analysis to the manuscript as Supplementary Figure 8 and Supplementary Table 9. We added on page 10, line 267ff: “An age-decade stratified analysis of the entire cohort, including adults and pediatric patients, indicated that accuracy increased from childhood to adulthood and remains stable through older adulthood (Supplementary Figure 8 and Supplementary Table 9).” and copied the results here for your convenience:

Age (years)	ALL	AML	APL
0–9	0,65	0,56	n/a
10–19	0,74	0,69	0,87
20–29	0,74	0,78	0,89
30–39	0,78	0,77	0,91
40–49	0,80	0,81	0,93
50–59	0,80	0,83	0,93
60–69	0,75	0,76	0,89
70–79	0,75	0,79	0,94
80–89	n/a	0,78	0,85

Abbreviations: 'n/a', no available AUC for that age/disease subgroup combination

4. The discussion is the same length as the results and is too long. Please shorten to focus on the take away messages rather than summarizing the results.

Response: Thank you. The manuscript has been changed as suggested. We shortened the discussion on page 12f. lines 331 ff., 338ff., 358ff. and 362 ff.

Reviewer 2 report:

(Remarks to the Author):

Turki et al. have compiled an international multicentric dataset of routine laboratory parameters from a diverse group of over 6000 patients and used it to improve an approach introduced previously by Alcazer et al. in 2024 on a French cohort and develop a group of algorithms for subtyping acute leukemia. The topic responds to a relevant medical question, namely quickly and easily subtyping AML patients, which has implications on treatment and potentially survival of this patient population. They also address subtyping in pediatric populations. Overall, the paper is well written. While the machine learning approach is largely derived from previous work, the machine learning approach used is well documented and the code made available.

Response: Thank you very much for the positive evaluation of our work.

Apart from these strengths, there are a few points that the authors should address before the manuscript is ready for publication.

Major points:

1) Large part of the scientific value of this work justifying publication in a journal like Nature Communications lies in compiling a unique and diverse global dataset of leukemia patients, especially as the algorithmic strategy has been described previously. Hence, data accessibility to the scientific community is key when considering the scientific impact of the manuscript. The authors address this point by intending to publicly share the dataset via the HARMONY Alliance repository. However, they are very vague on the amount of data or centers included in the final deposit ("centers willing to share their data"). For high relevance in the community, the anonymized dataset should be deposited in full or with minimal redactions in order to be accessible for independent testing and as a basis for future work. The authors should clearly state what they are willing to share in a revised version of the manuscript.

Response: Indeed, data availability for further research is an important contribution to the advancement of science and for the academic community, hence we included this statement. With the revised manuscript, we have signed a data sharing agreement contract with the HARMONY Alliance Foundation, a copy of which has been provided to the handling editor. We confirm that all investigators from the 20 study centers have agreed on this sharing policy (n=6,206 cases). Data is available from the HARMONY Alliance repository with the date of publication of this manuscript, as documented in the data availability statement.

2) The authors' outlier filtering method is at the heart of their refinement, but its description is unclear. In the relevant section, the authors state "Overall, this pipeline

identified incorrectly predicted samples with twice as high a probability as correctly predicted samples (54.79% vs. 24.46%), improving the algorithm's accuracy while including as many patients as possible for accurate predictions." It is not clear what the prediction then is for these cases. Are they excluded from the analysis? Also, could there be any pattern shared by the outliers removed that is not captured by the algorithm (e.g. secondary diagnoses or lab errors)?

Response: The outlier detector (i.e., cases identified as being out of distribution) is part of the pipeline improvement to ensure robust and generalizable predictions in patients with acute leukemia and to limit false-positive findings in patients presenting with leukocytosis or cytopenia, but without acute leukemia. Please note that the advantage of the outlier prediction is a) that it removes comparatively few samples and that b) it enables to use the algorithm safely on unknown samples with a high level of confidence. We believe that this trade-off is favorable: we exclude predictions on atypical or low-quality data to improve the reliability and interpretability of data within the learned manifold.

We followed the suggestion of Reviewer 2 and analyzed the removed outliers with a distribution analysis. Two patterns are noted:

- **Feature completeness has a limited impact. Outliers have a slightly higher proportion of missing features for several key inputs compared to the non-outliers. For instance, lymphocytes (+5.94%), monocytes (+5.52%), PT (+2.79%), WBC (+2.13%). Interestingly, fibrinogen has slightly lower missingness in outliers (-1.86%).**
- **As for the accuracy metrics, the proportion of detected outliers varies across centers. Low rates were detected across continents, e.g., Antananarivo (7.3%), Bochum (9.6%), Melbourne (11.3%), and Maastricht (12.2%). Very high outlier rates were more likely in centers with limited performance than in those with very high accuracy metrics, e.g., Lagos (82.7%), Kolkata (57.1%), Wroclaw (55.2%). Rome and Suzhou were around 33%. This pattern could be leveraged for benchmarking purposes.**

Overall, these differences indicate that data quality and completeness contribute to the "outlier" signal and sample exclusion.

3) Before model refinement, there is considerable inter-site difference for the dataset, down to models that do not work at all (AUROC 0.5) in some cases, as shown in Fig. 2. Can the authors provide this full analysis also for the post-refinement model? This would provide a more comprehensive overview of the site-specific impact of their refinement method.

Response: As requested, we added the per-center analysis of the centrally tested cohort after refinement as Supplementary Figure 6. Across centers, most accuracy metrics improved after refinement (e.g., AUC AML in Barcelona 0.905 (+0.04);

Salamanca 0.987 (+0.12); Suzhou AML 0.941(+0.11)). Missing “after” AUROC entries indicate that no evaluable samples remained post-refinement for that site–class. Even in sites with previous poor performance, e.g. Kolkata, high accuracy predictions were obtained for ALL 0.91 and AML 0.9. The adequate performance of APL prediction in the Lagos cohort (AUC APL 0.92) is contrasted with its poor predictions in AML, which remained unchanged (AUC 0.58). However, the Lagos cohort had substantially distinct feature distribution and leukopenia, was numerically very small and had the highest proportion of missing features beyond the level that we accepted for all other centers, as coagulation features were systematically missing on the African cohorts. Please note that the AUC curve of Dallas is technically skewed since only 4 APL samples and no ALL samples were provided into the study. Its data however populated the overall study population and provided highly accurate predictions for AML as documented by the other metrics. We copied the results here for your convenience:

4) While the authors achieve a substantial improvement using their outlier removal strategy, classification performance based on lab parameters is still far from the diagnostic gold standard. What strategies to the authors foresee to improve from this point? Do they believe that further increasing case numbers or including other clinical values would be important?

Response: We think that this comment refers to the discussion of accessibility versus precision. The main question is what the algorithm will be used for. Indeed, it remains behind the current gold standard of leukemia care in high-income countries, but that standard includes bone marrow examination with cytogenetics as well as molecular genetics and flow cytometry to achieve its high diagnostic accuracy. However, this precision comes with high costs and still requires time today. While most recent research (Steinicke T et al., *Nature Genetics*, 2025; Marchi F et al., *Nature Communications*, 2025) is addressing this issue using long read sequencing for rapid diagnosis, the infrastructure implementation remains costly. This is also true for processing, even though individual samples sequencing is reported to cost several hundred dollars (Furlan S, *The Hematologist*, 22 (6), 2025) instead of thousands, this remains above the affordable level for leukemia screening in LMIC. In contrast, the algorithm and pipeline reported in this manuscript use standard laboratory which are accessible, cheap and provide an immediate answer, which in our view may support early triage and support referral, particularly in resource poor settings. The results that we provided with the revised manuscript further support the validity of the pipeline, including the out-of-distribution detector. With the over 6,000 cases that we have readily assembled, we do not think that performance will be substantially increased by further adding patients for training. Yet, the variation in the features and exploration of additional features appears promising. As leukemia is a disease of the marrow cells, its examination remains the ultimate diagnostic tool. Approaches that first foresee a peripheral blood examination for triage and a marrow examination as second step have been followed in Hematology for decades and today may both be AI-supported, and, depending on the resources, can be combined with (epi-)genetics data for full diagnosis.

Minor points:

1) When considering pediatric cases, the authors find that different relevant lab value constellations differ from the adult case. Do the authors believe that this reflects the distinct biology of pediatric disease, or is it due to the different case number distribution

Response: Indeed, several factors may explain the observed differences, including case number distributions and distinct biology, which likely both contribute to the

difference between groups. We followed the suggestion of Reviewer 2 and performed a downsampling experiment with the adult cohort (n=3979) to match their case numbers and disease proportions of leukemia subgroups (AML, APL and ALL) to those of the pediatric cohort (n=1414). The downsampling decreased the model's AUC (e.g. adult AML from 0,82 to 0,77, but it remained above the range of the pediatric cohorts (AML AUC 0,64). This finding supports the hypothesis that both factors contribute and that the case number distribution accounts for about 2-6% of this AUC difference, depending on the leukemia subtype, while the biology appears to contribute to the remaining 2-13%, with the greatest difference observed for AML. With respect to biological differences there is an extensive repertoire of publications (e.g. Bolouri H et al. *Nature Medicine* 2018, Roberts KG *Am Soc Hematol Educ Program*, 2018). The results of our experiments are shown below:

Accuracy metrics:

Pediatric cohort metrics (n=1,414)

Metric	ALL	AML	APL

AUC	0.7436	0.6381	0.8093
Accuracy	0.6198	0.6438	0.9435
Precision	0.9409	0.1562	0.1026
Recall	0.6093	0.5734	0.4444
F1 Score	0.7396	0.2455	0.1667

Adult cohort metrics (n= 3,979, original, centrally tested)

Metric	ALL	AML	APL

AUC	0.7924	0.8169	0.9209
Accuracy	0.8223	0.7851	0.9256
Precision	0.4551	0.8433	0.7116
Recall	0.4783	0.8663	0.5267
F1 Score	0.4664	0.8547	0.6053

Adult cohort metrics (n= 1,414, proportions matched to kids)

Metric	ALL	AML	APL
AUC	0.7721	0.7745	0.8649
Accuracy	0.5237	0.5661	0.9406
Precision	0.9723	0.1715	0.1071
Recall	0.4761	0.8601	0.5000
F1 Score	0.6392	0.2860	0.1765

2) Presentation: Figures are overall quite busy, with small legends, especially in Fig. 2 and 3.

Response: The manuscript legends of Figure 2 have been revised as suggested. We also have updated Figures 3 and 5 including minor adjustments.

3) In Fig.2, the same formatting should be used for all graphs, and multiple similar legends avoided.

Response: The manuscript Figure 2 has been changed accordingly.

Reviewer 3 report:

(Remarks to the Author):

In this manuscript, the authors have validated and improved upon a published model for the prediction of acute leukemia subtype across several centres. They use CBC counts, blood LDH levels, and coagulation parameters to predict the leukemia subtypes. The concept, overall, is promising and tries to solve an important problem, i.e., lack of access to sophisticated investigations in developing countries.

Response: Thank you very much for the positive evaluation of our work.

That being appreciated, these are the concerns that are noted:

1. Patients with acute leukemia (AL) can have varied clinical presentations from frankly

leukemic (elevated WBC counts) to cytopenic. Frank leukemic presentations are also noted in other myeloid malignancies such as chronic myeloid leukemia and other MPNs, as well as MDS-MPN overlap conditions, such as chronic myelomonocytic leukemia in adults and juvenile myelomonocytic leukemia in children, which have monocytosis. These presentations are also seen in lymphomas such as chronic lymphocytic leukemia or mantle cell lymphoma, to name a few. Similarly, AL also commonly presents with frank cytopenia mimicking other malignancies, for example, MDS, CMML (dysplastic type), and reactive conditions (eg, B12/Folate deficiency, sepsis, or chemotherapy-induced cytopenia). The training set that is used for the model uses features of frankly leukemic patients and, in all probability, excludes patients with other conditions or leukemia that have been partly treated with disease-modifying therapy (as stated in their Lancet Digital Health paper). This is a major shortcoming of the study, as these are common problems in the real world and are amplified in LMIC settings, which is the premise of this paper.

Response: We agree with Reviewer 3 that differential diagnoses of acute leukemia are a challenge in clinical care and have acknowledged this potential limitation in the previously submitted manuscript. In response to diagnostic challenges, the version already included a preprocessing pipeline for the identification of cases that were out of distribution.

Following the suggestions of Reviewer 3, we further expanded the preprocessing pipeline. We validated it with several cohorts of patients with possible differential diagnoses to acute leukemia (n=341) presenting with leukocytes above, within and below the normal range. It included frank leukemic presentations (e.g. chronic lymphocytic leukemia, mantle cell lymphoma, chronic myeloid leukemia, or myeloproliferative neoplasms) or cytopenic conditions (including a variety of diseases, e.g., MDS, MPN, B12 deficiency. We also had a few patients with Juvenile myelomonocytic leukemia. Laboratory results of patients with severe reactive conditions, in particular with SARS-CoV-2 and sepsis, were obtained from two public datasets (Ordoñez-Avila R, et al. *Data Brief*. 2023, and Johnson, A. et al. *Sci Data*, 2016). The updated preprocessing pipeline including filters (e.g. proportion of neutrophils in leukocytosis and cytopenia) assessment and the out of distribution detector was tested with these patients. The proportion of misclassified patients varied depending on the constellation: Patients with leukocytes above the normal were well identified as either acute leukemia or having an out of distribution constellation (87.9%, Supplementary Table 8) using only laboratory data. In cytopenic conditions, the rates were lower (78.4%), as expected, because the differentiation of some differential diagnoses to acute leukemia can be challenging with the CBC alone. For instance, MDS and AML cases may have a broad overlap in peripheral blood phenotypes and mutated genes. In particular in formerly AML-MR, according to the current ICC classification (Arber et al. *Blood* 2022). Dysplasia in marrow smears and histology and the percentage of marrow blasts support the diagnosis (Arber et al. *Blood* 2022, Khoury et al. *Leukemia* 2022). However, there is also evidence that the blast count may be substantially overlapping between both

entities (Estey et al. *Blood* 2022). For high-risk MDS with adverse genetic settings, this ambiguity may translate into similar immediate care measures in the emergency room and to a therapeutic repertoire involving early consolidating HSCT in eligible patients.

We think that combining regular medical practice (including anamnesis, physical examination, and initial emergency unit assessment) with the calculator's results should remain the approach to these cases at the edge of secondary AML. We copied the results for your convenience:

Category	Out of distribution, %	Out of distribution, n	Total, n
Leukocytes below normal	78.4%	40	51
Leukocytes above normal	87.9%	109	124
Leukocytes within normal	90.4%	150	166
Diagnosis subsets			
Malignancies			
Chronic lymphocytic leukemia	81.3%	26	32
Chronic myeloid leukemia	94.7%	18	19
Juvenile myelomonocytic leukemia	100.0%	2	2
Lymphoma	62.5%	5	8
Myelodysplastic syndromes	60.0%	6	10
Myeloproliferative neoplasms	87.5%	7	8
Non-malignant disorders			
Vitamin B12 deficiency	71.4%	10	14
SARS-CoV2	93.7%	118	126
Sepsis	90.4%	103	114

Others: Individual cases (each n= 1) with folate deficiency, PNH, TTP, parvovirus B19 infection and EBV infection were not reported as separate entities but are included in the overall statistics.

2. Validation studies to refute the diagnosis of acute leukemia: This ML model should be validated on conditions that are not acute leukemia but present with leucocytosis and monocytosis (lymphoma/chronic lymphoproliferative disorders, MDS-MPN overlap syndromes), as well as cytopenia such as MDS, MDS-MPN, and acute leukemia with cytopenic counts. This will ensure that this model has true clinical value and is applicable to a resource constrained setting.

Response: Although this was not our initial study design, we followed this suggestion to validate the preprocessing pipeline and the model in conditions presenting with leukocytosis and monocytosis, as well as in those with cytopenia. The results

presented above indicate that the preprocessing pipeline has a low rate of false positive findings in cases with leukocytosis, which in our view have the greatest need for rapid action, and that the classifier using limited features has indeed clinical value in this constellation. Such a classifier may potentially accelerate the initiation of cytoreductive treatments in patients with acute leukemia depending of their myeloid (AML) or lymphoid (ALL) lineage in resource constrained settings. We added the following to the results section on page 9, line 243: "The developed preprocessing pipeline did also allow to challenge the algorithm with differential diagnoses to AML, ALL and APL presenting with leukocytosis or cytopenia (Supplementary Table 8). Cases with leukocytes above the normal were excluded with high accuracy (87.9%), while cytopenic cases were more challenging (78.4% of leukopenia cases with other diagnoses excluded)."

Beyond LMIC, we think that this tool may also be beneficial for triage purposes in countries with high income or as a referral tool in resource-poor settings.

3. It is unclear how patients were selected for training the model or for validation. Are these consecutively diagnosed patients, or is there a selection bias?

Response: Participating centers were contacted in the second half of 2024 and invited to submit consecutive patients. No selection was applied except for data availability at diagnosis and completeness of data at diagnosis. The probability of selection bias is therefore low. Plausibility checks and manual data review was centrally performed but individual centers were not externally monitored as in clinical trials.

4. There are minor errors in the supplementary data as total WBC, monocyte, platelet counts are in G/L

Response: Thank you, there was an inconsistency, which is now solved. While the representation of CBC results in Germany and the UK is common in counts per nanoliter e.g. leukocytes of 4.0 - 11.0 x 10⁹/L, representations in g/L are comparable and have also been adopted by Alcazer et al. *Lancet Digit. Health* 2024. We changed the supplementary information into g/L.

Reviewer #3 (Remarks on code availability):

Yes, the github repository does have info on how to run the pipeline and generate synthetic data

<https://github.com/UMEssen/aipal-validation>

Point by point response to peer review for “International testing and refinement of AI algorithms predicting acute leukemia subtypes from routine laboratory data” by Turki et al.

Remarks to the Author

Reviewer 1: The authors have satisfactorily addressed my comments.

Response: We thank the reviewer for the positive evaluation of our work.

Reviewer 3: The authors have answered all queries. no further comments

Response: We thank the reviewer for the positive evaluation of our work.

Reviewer 4: The authors have adequately addressed all the points raised. However, several aspects of the abstract remain unclear and would benefit from clarification:

L89: The authors first report the performance of the “confident” AI-PAL predictions (AUROC 0.94/0.98/0.84 for acute myeloid leukemia/acute promyelocytic leukemia/acute lymphoblastic leukemia). Are these results derived from the overall merged dataset, or from a specific cohort or data split (e.g., test set)?

L93: The AUROC reported for acute myeloid leukemia is 0.72 here, whereas 0.94 is reported immediately before. Since the test set is not defined in the abstract, it is difficult to interpret what this comparison refers to (e.g., random split of the pooled cohort, external dataset, or a selected subset).

I therefore suggest a minor revision of the abstract to clearly specify what is being referred to in terms of the baseline model and the evaluated cohorts.

Response: We thank the reviewer for the positive evaluation of our work.

Hereby, we clarify the points raised by reviewer 4 and have adjusted the abstract accordingly. The high AUROC values in the abstract (0.94 in AML, 0.98 in APL and 0.84 in ALL) were obtained by applying the previously published confidence cutoff to the pooled adult cohort, which retains only predictions with high model certainty. However, this cutoff excluded 70.8–92.5% of patients from predictions, limiting its clinical applicability on many patient cases or leaving uncertainty for those patients not qualified for confident predictions. We therefore focused on the baseline model without cutoff. The baseline AUROC of 0.72 for AML shown in the abstract refers to the algorithm performance on the hold-out test set (n=2,692) used for evaluating the outlier detection refinement pipeline that we present in the manuscript. This test set comprised all patients whose predictions were below the 0.9 confidence threshold. Our refined approach improved the AUROC from 0.72 to 0.84 on this challenging subset while excluding only 12.1% of patients from predictions. We have shortened the abstract as requested by the editorial office and specified the evaluated cohorts and the rationale for comparing these metrics as requested by Reviewer 4 and copy it here for your convenience. The tracked version is provided within the manuscript.

Despite advances for patients with acute leukemia health disparities limit access to diagnosis and treatment. Artificial Intelligence (AI) approaches may address some disparities. We retrospectively assembled a diverse, international cohort of 6,206 leukemia patients from 20 centers to test an AI tool designed to support leukemia diagnosis using standard laboratory results. Executing the pretrained algorithm resulted in varying accuracy metrics. With confidence cutoff predictions, 2000-fold bootstrapped area under the curve (AUROC) metrics were 0.94 for acute myeloid leukemia (AML), 0.98 for the promyelocytic subtype and 0.84 for acute lymphoblastic leukemia. However, this cutoff excluded 70.8–92.5% of patients from predictions. We improved accuracy and robustness, while maintaining generalizability via an ensemble of Isolation Forest and Local Outlier Factor increasing AUROC for AML from 0.72 to 0.84 (hold-out test set, patients below confidence threshold), while excluding only 12.1% of patients. Furthermore, we retrained the algorithm for pediatric patients.